# A DDoS Attack Detection Method Using Conditional Entropy Based on SDN Traffic

**Qiwen Tian \* and Sumiko Miyata**

Department of Electrical Engineering and Computer Science, Shibaura Institute of Techonology, 3-7-5 Toyosu, Koto-ku, Tokyo 135-8548, Japan
\* Correspondence: ma21099@shibaura-it.ac.jp

**Abstract:** To detect each network attack in an SDN environment, an attack detection method is proposed based on an analysis of the features of the attack and the change in entropy of each parameter. Entropy is a parameter used in information theory to express a certain degree of order. However, with the increasing complexity of networks and the diversity of attack types, existing studies use a single entropy, which does not discriminate correctly between attacks and normal traffic and may lead to false positives. In this paper, we propose new state determination standards that use the normal distribution characteristics of the entropy value at the time which an attack did not occur, subdivide the normal and abnormal range represented by the entropy value, improving the accuracy of attack determination. Furthermore, we show the effectiveness of the proposed method by numerical analysis.

**Keywords:** entropy; SDN; attack detection; DDoS; abnormal traffic; flash crowds

## 1. Introduction

In recent years, as sensors become smaller and more energy efficient and networks become more diverse, the Internet of Things (IoT) [1] is being more widely used in various fields. As a result of connecting various "things" to the Internet, traffic increases due to the transmission of real-time data and status confirmation messages, making the composition of the network increasingly complex. Therefore, flexibility in various network environments is a challenge in designing new network architectures [2]. Conventional networks cannot be centrally managed by administrators because routers, switches, firewalls, etc., must be changed individually when servers or network devices are added or the network configuration is changed. To solve these problems, the concept of an SDN (software-defined network), "network virtualization," in which the entire network is controlled by software, has become popular [3]. The most prominent feature of an SDN is the separation of the data and control planes, i.e., the "transmission function" and the "control function" are implemented by different network devices. Therefore, network devices in the data plane do not send messages to each other to share the network status but only forward packets. On the other hand, the SDN controller in the control plane manages the entire network and can quickly change and update network settings according to traffic conditions. By leveraging the characteristics of an SDN network, an IoT environment with complex networks such as sensor networks, wireless networks, etc., can also achieve unified network management [4].

However, one of the most pressing concerns in the realm of IoT security is the threat of Distributed Denial of Service (DDoS) attacks, which can cause significant disruption to IoT networks and systems. DDoS attacks involve overwhelming a network with traffic from multiple sources, rendering it inaccessible and disrupting normal operations. The potential consequences of a successful DDoS attack on an IoT network can be severe, ranging from data loss and theft to physical harm and even loss of life in critical infrastructure environments such as healthcare, transportation, and energy. The impact of these attacks can be amplified by the sheer scale and complexity of IoT networks, which often involve

multiple devices and layers of connectivity. Moreover, many IoT devices are designed with limited security features, making them easy targets for attackers. To this end, a growing body of research is focused on developing new security mechanisms and techniques that can protect IoT networks and systems from DDoS attacks. These include approaches such as machine learning-based anomaly detection, intrusion detection and prevention, and network-based defense mechanisms.

Additionally, in the SDN environment, with the separation of the "transmission function" and the "control function", attacks on the control plane will become even more valuable than those on the data plane, which merely transfers data [5]. Attackers can target the SDN controller and send a large number of request messages from data plane devices to make the SDN controller inoperable. Therefore, it is necessary to detect attacks not only by considering attack methods that have been used in conventional architectures but by also considering new attack methods that have been introduced in SDN architectures.

The IDS (intrusion detection system) [6] is a system that detects attacks such as those described above. In general, an IDS collects traffic information such as IP addresses and port numbers in advance for intrusion detection. It then defines attack patterns, which are traffic sets that match attack communications, and normal patterns, which are traffic sets that match normal communications. For example, Figure 1 assumes that users and servers communicate via an external network. In this case, the green arrows represent the traffic between the server and user, and the IDS acquires and analyses this traffic to determine normal and abnormal communication.

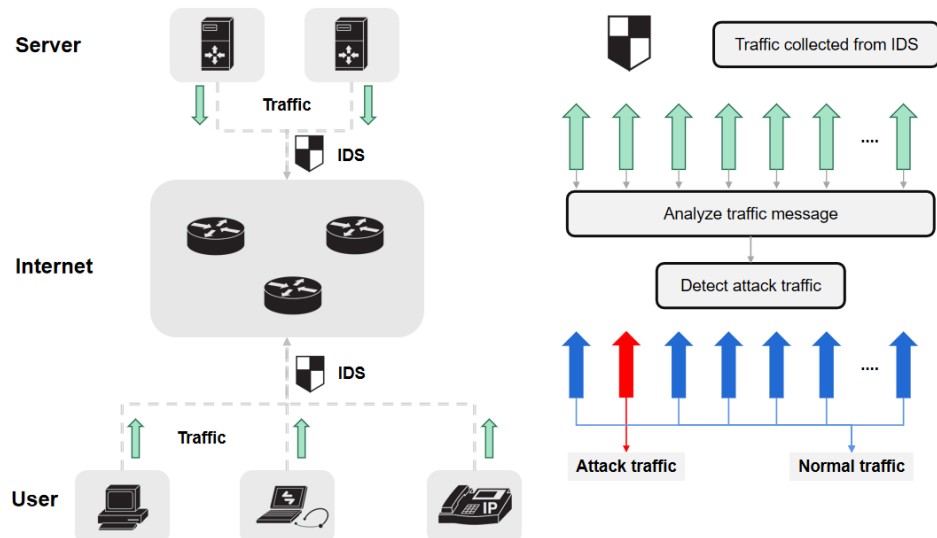

**Figure 1.** Intrusion detection system.

As shown in Figure 2, an IDS can be broadly classified into two types: the signature type, in which a new communication is judged on the basis of whether it matches an attack pattern; and the anomaly type, in which it is judged on the basis of whether it matches a normal pattern [7]. Each red arrow represents an attack pattern extracted from conventional attacks on the system, and the blue arrows are the ranges determined by referring to normal traffic. In other words, the signature type judges traffic to be an attack communication when it matches the red attack pattern in the figure. The anomaly type defines a normal pattern and determines an attack to be occurring when traffic that differs from the normal range occurs. Since the anomaly type has the advantage of being able to detect unknown attacks outside the normal range, many studies have been conducted [8–10].

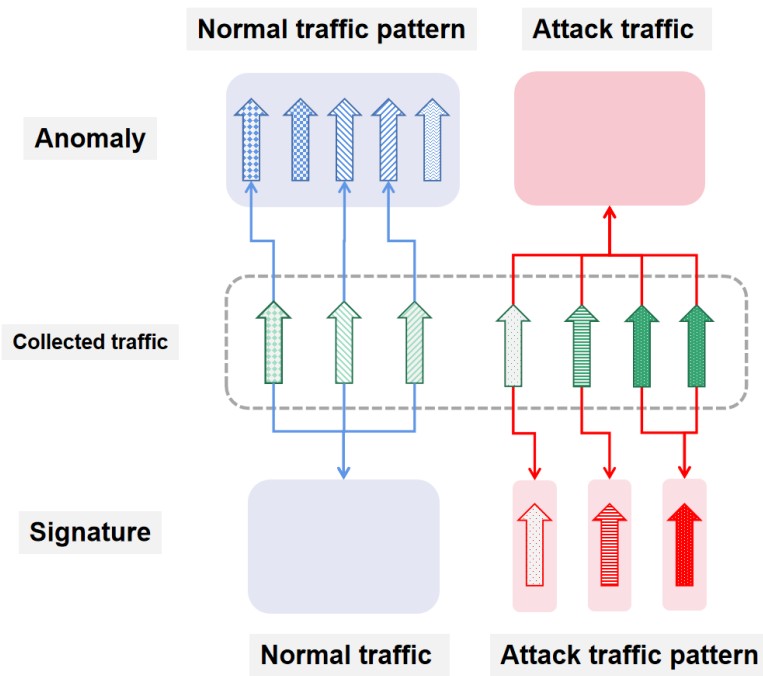

**Figure 2.** Anomaly-type and signature-type IDS.

However, previous entropy methods [9], when implemented in SDN environments, should also consider attacks on the control plane on the basis of the SDN architecture features. However, there are only entropy detection methods for flooding attacks in existing methods.

The normal traffic range in the anomaly type is also important. As shown in Figure 3, false positives can occur if the normal traffic range is too large or too small compared with the actual normal traffic range. For example, when an event is held, there could be a network failure due to a flash crowd, which is a rapidly increasing number of accesses [11]. In such a case, the traffic may be considered to be attack traffic even though it is normal traffic. In other words, to apply the entropy method to an SDN, the range of normal traffic should be re-examined after fully considering the changes in the network structure caused by the SDN.

In this study, we propose an attack detection method using conditional entropy based on the anomaly-based entropy method, which takes into account the changes in the network structure due to an SDN and multiple traffic factors, such as flash crowds. In addition, by focusing on traffic outside the set normal traffic range, such as a flash crowd, we use a signature-based detection method to define the normal traffic range by determining the type of normal traffic and attack traffic with similar characteristics.

The contributions of this study are summarized below.

- Taking into account the practical implementation of the SDN architecture, this study extends the recognition of packet-in attacks and flash crowds on the basis of prior research, thereby reducing false positives.

    – For different attack strengths, this method can make accurate judgment of the attack type possible and facilitate the subsequent targeted treatment of the attack.

- By utilizing conditional entropy in conjunction with the existing method, the parameters for evaluating normal flows are expanded, thereby enhancing the detection accuracy.

    – The accuracy in judging traffic with similar characteristics is improved on the basis of existing methods.

-   The detection methods of the anomaly and signature types are combined to ensure that the difference between abnormal traffic with similar characteristics can be determined, while also providing protection against attacks that have not yet been identified.

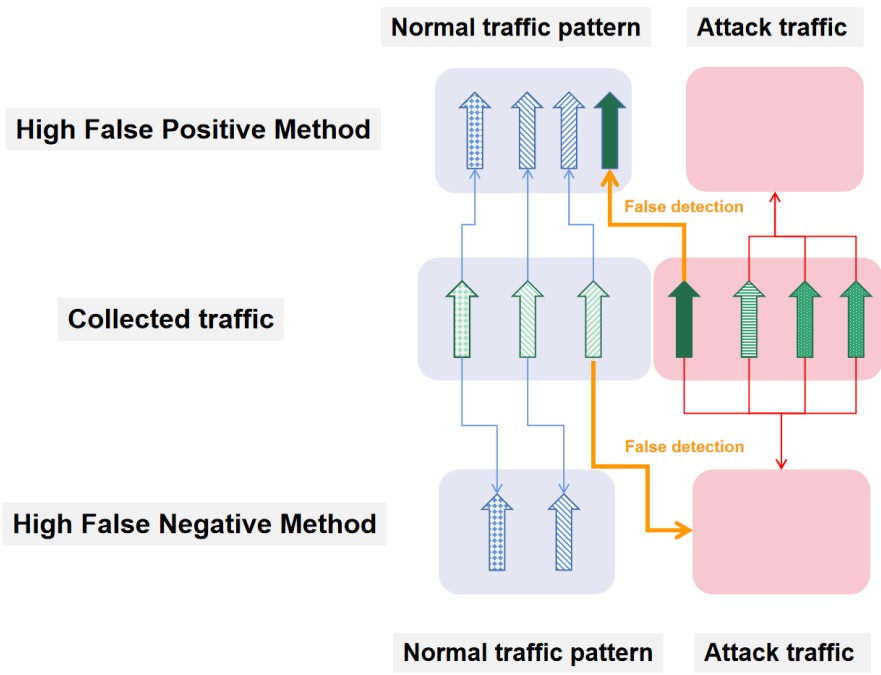

**Figure 3.** False detection in an IDS of anomaly type.

The remainder of this paper is organized as follows. Section 2 describes related research in DDoS attack detection, and Section 3 defines conditional entropy and describes the data processing procedure for the proposed method. Section 4 presents a numerical analysis, and Section 5 summarizes and discusses problems and issues in this study.

## 2. Related Works

As described in the previous section, detection methods can be divided into two types: signature-type methods [12,13], which determine an attack by matching the attack traffic pattern; and anomaly-type methods [8–10,14,15], which determine an attack by matching the normal traffic pattern.

In the former attack detection method, attack patterns are registered one by one in advance, and when attack traffic occurs, the signatures of the predefined attack patterns are checked to determine whether the traffic is an attack or not [16]. The advantage of this method is that it is based on attack patterns that have been experienced in the past, and control can be achieved through clarified characteristics. However, it has the disadvantage that when a new attack occurs, it is judged as normal because it does not match an attack signature. An attack traffic list needs to be stored, updated, and maintained, which increases the processing load. P. Ioulianou et al. [12] proposed signature-based intrusion detection to deal with two variations of DDoS attacks. The simulation outcomes demonstrated that these attacks may affect the reachability of specific IoT devices and their energy consumption. M. Aldwairi et al. [13] proposed facing the problem of significant overheads due to memory usage and execution time due to the signature pattern matching operation by parallelizing the process on a multi-core CPU.

The latter method defines a normal pattern and determines whether or not a pattern is an attack. This method determines illegal communication by comparing traffic with the defined pattern. In other words, it is not necessary to include all attacks; unknown attacks can be detected after defining normal patterns, which is an advantage of the anomaly-type method. On the other hand, if there is a mistake in the defini-

tion of the normal pattern, or if anomalous traffic similar to the normal pattern occurs, false positives are likely to occur. Currently, many studies are focusing on the anomaly-type method and using statistical, neural network, machine learning, and other methods to reduce the false positive rate while consolidating the range of normal patterns [17]. Kyaw et al. [14] proposed a method that increases the detection rate of DoS flooding attacks by switching to a polynomial SVM algorithm over the traditional linear SVM algorithm. Liu et al. [15] proposed a method using a CGAN neural network algorithm focusing on fast flooding attacks. Carvalho et al. [9] computed entropy using a target IP as the only tick. They proposed a method for defining a normal pattern with reference to the change in entropy. However, since SDNs have more attack patterns than conventional networks, the probability of false positives increases if the decision is based solely on the target IP.

Based on the existing entropy method [9], we introduce conditional entropy based not only on the target IP but also on the actual attack features, so that we can understand the changes of the entire network in a multidimensional manner. In terms of defining patterns, we use anomaly-type methods that define normal patterns among all traffic in combination with signature-type methods that define similar patterns for each of them, in order to avoid false positives while maintaining detection rates based on the diversity of normal patterns.

## 3. Proposed Method

### 3.1. Assumed Environment

#### 3.1.1. SDN Environment

In SDNs, two types of planes are used to separate data transfer and control functions: a data plane for data transfer and a control plane for centralized management. The data plane is the part of the network equipment that performs the data forwarding process. Unlike conventional network devices, the OpenFlow switch in an SDN only accepts routing information from the SDN controller; thus, it only searches for the destination of received packets in a table, determines the destination, and forwards the packet. If the destination does not exist in the table, the OpenFlow switch sends a packet-in message to the SDN controller saying that a new packet is arriving, and the SDN controller decides how to process the packet before sending another message to the OpenFlow switch informing the switch of the processing method for the packet [18].

On the other hand, the control plane controls the data plane. It can centrally control the entire network and plays the role of creating and controlling routing information, such as routing tables and MAC address tables, necessary for forwarding data [19].

For these reasons, both the SDN controller and OpenFlow switch are indispensable for building an SDN. However, in an SDN, since the SDN controller centrally manages the directed traffic, attacks on the control plane belonging to the SDN controller are more valuable from an attacker's point of view [20].

#### 3.1.2. DDoS Attacks and Other Anomalous Traffic

Before explaining the proposed attack detection method, we first explain the working principle and characteristics of DDoS attacks. A DDoS attack is an attacker launching a simultaneous attack on a specific server using multiple devices. The goal of this attack is to abnormally increase traffic and stop server functions [21]. Since the number of attacker devices is larger than that of conventional DoS attacks, the damage is larger than that of conventional DoS attacks. Specifically, the name of the attack can be classified as SYN ACK flooding, UDP flooding, or ICMP flooding depending on "which part" of the mechanism the attack uses to send messages [22].

In particular, a new DDoS attack called a packet-in attack exists against the SDN architecture focused on in this study [23]. The packet-in attack is an attack that increases the number of packet-in messages sent to the SDN controller by the OpenFlow switch, which cannot refer to a routing table, by sending a large number of fake packets to an IP for which the attacker does not exist. The purpose of this attack is to cause the controller to lose control of the SDN network by abnormally increasing the number of packet-in messages.

On the other hand, flash crowds [24], which are similar to DDoS attacks, are accesses to servers caused by events that increase people's interest, such as bargain sales and the World Cup. For example, the number of user accesses to a particular web site may increase so much in a short period of time that some users are denied access as legitimate [25]. The result is the same as a DDoS attack, despite it not being an attack.

*3.2. DDoS Detection Method Using Conditional Entropy*

3.2.1. Disorder State Based Attack Detection

In this research, we propose an attack detection method based on the "state of disorder". Here, "state of disorder" means whether the traffic parameters are random or not during a certain time period, as shown in Figure 4. For example, if parameters such as the destination IP and source IP of packets generated during a certain time period are concentrated at the same value, an attack may be occurring [26]. We introduce entropy to quantify the degree of concentration.

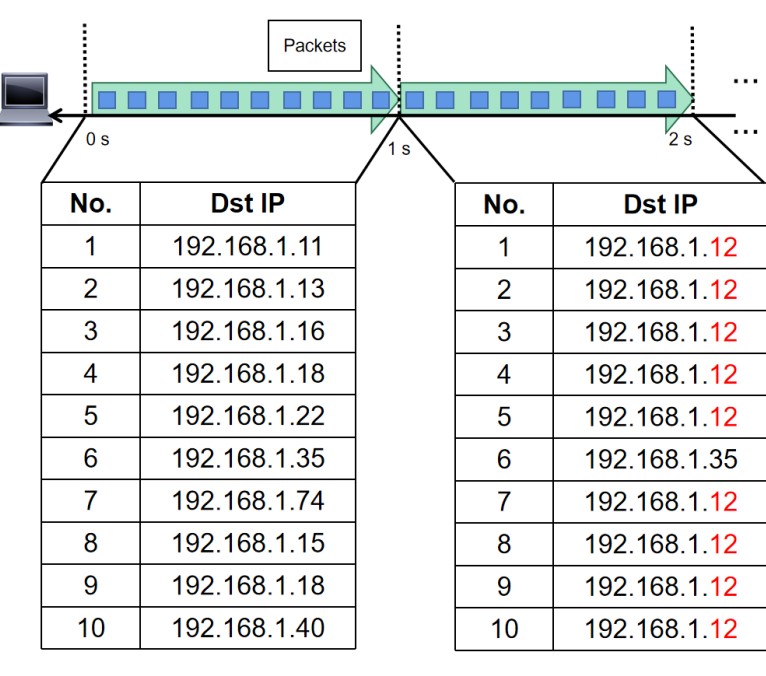

**Figure 4.** Randomness of a single parameter.

In a normal usage environment, there exists a certain correlation between various parameters of data packets based on user operations and business requests, while a single entropy can reflect the degree of dispersion of a single parameter, it cannot represent the overall situation of multiple parameters. As depicted in Figure 5, the occurrence frequency of destination IP addresses and source ports for 10 packets within 1 and 2 s are the same, and calculating the single entropy of the two parameters will yield identical results. However, it cannot indicate the complete consistency of the two parameters in combination. Therefore, in this study, we introduced conditional entropy as a measure of the uncertainty of multiple parameters as a whole.

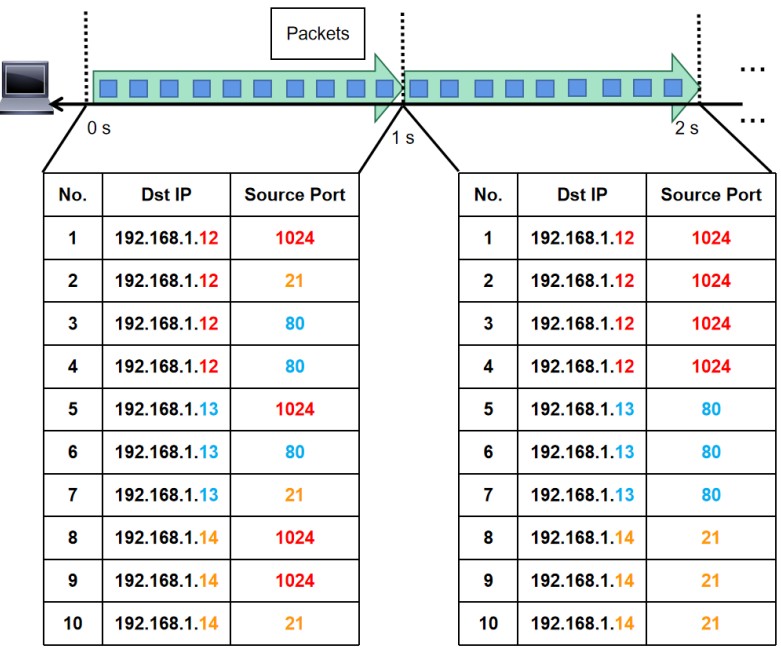

**Figure 5.** Randomness of complex parameters.

### 3.2.2. Entropy and Conditional Entropy

In this study, we use entropy and conditional entropy together to quantify the degree of concentration of each parameter of the traffic at the observed time of $T$ seconds. As shown in Algorithm 1, first, the collected packets are sorted in order of arrival, and these data are divided into $N$ pieces. Let $\mathcal{W} = \{\mathcal{W}_1, \mathcal{W}_2 \ldots, \mathcal{W}_N\}$ be the set of these divided data. For each $\mathcal{W}_i, i \in \mathcal{N} = \{1, 2, \ldots, N\}$ of this dataset, $m$ packets are randomly selected, and the selected set is redefined as window $\mathcal{W}_i = \{a_1^i, a_2^i, \ldots, a_m^i\}$. In each window $\mathcal{W}_i$, the number $D_i$ of different IP addresses $d_j, 1 \le j \le D_i$ ,$S_i$ of different source port numbers $s_j, 1 \le j \le S_i$, $L_i$ and $L_i$ of different packet sizes$l_j, 1 \le j \le L_i$ are expressed in the form of set $\mathcal{D}_i$, $\mathcal{S}_i$, and $\mathcal{L}_i$.

The probability $p_{i,j}^{\mathrm{di}}$ of the occurrence of the target IP $d_j$ in each window $\mathcal{W}_i$ is calculated as follows, and $f_j^{\mathrm{di}}$ is the number of occurrences of each objective IP in window $i$.

$$p_{i,j}^{\mathrm{di}} = \frac{f_j^{\mathrm{di}}}{m}, i \in \mathcal{N}, j \in \mathcal{D}_i \tag{1}$$

Then, using the probability $p_{i,j}^{\mathrm{di}}, 1 \le j \le D_i$ of the target IP, the entropy $H_i^{\mathrm{di}}, i \in \mathcal{N}$ of the target IP is calculated as follows.

$$H_i^{\mathrm{di}} = -\sum_{j=1}^{D_i} p_{i,j}^{\mathrm{di}} \log_2\left(p_{i,j}^{\mathrm{di}}\right) \tag{2}$$

To clarify the type of abnormal traffic, the conditional probability $p_{i,j,k}^{\mathrm{sp}}$, $p_{i,j,n}^{\mathrm{ps}}$ of the source port and packet size for a given destination IP is calculated using the following equation with condition $1 \le k \le S_i, 1 \le n \le L_i, 1 \le j \le D_i$.

$$p_{i,j,k}^{\mathrm{sp}} := \mathrm{P}(s_k|d_j), s_k \in \mathcal{S}_i, d_j \in \mathcal{D}_i \tag{3}$$

$$p_{i,j,n}^{\mathrm{ps}} := \mathrm{P}(l_n|d_j), l_n \in \mathcal{L}_i, d_j \in \mathcal{D}_i \tag{4}$$

Using the obtained conditional probabilities, the conditional entropy of the source port and packet size for a given destination IP is obtained by the following equation.

$$H_i^{\text{sp}} = -\sum_{j=1}^{D_i} p_{i,j}^{\text{di}} \sum_{k=1}^{S_i} p_{i,j,k}^{\text{sp}} \log_2\left(p_{i,j,k}^{\text{sp}}\right) i \in \mathcal{N} \tag{5}$$

$$H_i^{\text{ps}} = -\sum_{j=1}^{D_i} p_{i,j}^{\text{di}} \sum_{n=1}^{L_i} p_{i,j,n}^{\text{ps}} \log_2\left(p_{i,j,n}^{\text{ps}}\right) i \in \mathcal{N} \tag{6}$$

---

**Algorithm 1** Entropy calculation algorithm.

---

1: For the collected data packets, classify the data packets that occur at the same second and randomly select $m$ put into the window $\mathcal{W}_i, i \in \mathcal{N}$.
2: Classify the different destination IP addresses, source port, and packet size in set $\mathcal{W}_i$ as $\mathcal{D}_i, \mathcal{S}_i, \mathcal{L}_i$.
3: Calculate the probability $p_j^{\text{di}}, p_{jk}^{\text{sp}}, p_{jn}^{\text{ps}}$ according to Equations (1), (3) and (4). Then, calculate the entropy whose output is $H_i^{\text{di}}, H_i^{\text{sp}}, H_i^{\text{ps}}$ according to Equations (2), (5) and (6).
4: These steps stop if the predefined loop $N$ is reached; otherwise, they return to step 2.

---

**Algorithm 2** Abnormal traffic determination algorithm.

---

**Require:** a set of entropy values for normal traffic $\mathcal{W}_t$, total traffic $\mathcal{W}_i$, $\mathcal{W}_t \subset \mathcal{W}_i, i, t \in \mathcal{N}$, average $\overline{H^{\text{di}}}, \overline{H^{\text{sp}}}, \overline{H^{\text{ps}}}$, standard deviation $\sigma^{\text{di}}, \sigma^{\text{sp}}, \sigma^{\text{ps}}$

**Ensure:** 0: attack, 1: normal

1: **for** each $\mathcal{W}_i, i \in \mathcal{N}$ **do**
2:  **if** $\overline{H^{\text{di}}} + 2\sigma^{\text{di}} < H_i^{\text{di}}$ **then**
3:   Traffic type $\leftarrow$ *Normal traffic*
4:   **return** 1
5:  **end if**
6:  **if** $\overline{H^{\text{di}}} - 2\sigma^{\text{di}} \leq H_i^{\text{di}} \leq \overline{H^{\text{di}}} + 2\sigma^{\text{di}}$ **then**
7:   Traffic type $\leftarrow$ *Normal traffic*
8:   **return** 1
9:  **end if**
10:  **if** $\overline{H^{\text{di}}} - 2\sigma^{\text{di}} > H_i^{\text{di}}$ **then**
11:   **if** $\overline{H^{\text{sp}}} + 2\sigma^{\text{sp}} < H_i^{\text{sp}}$ and $\overline{H^{\text{ps}}} + 2\sigma^{\text{ps}} < H_i^{\text{ps}}$ **then**
12:    Traffic type $\leftarrow$ *Flash Crowd*
13:    **return** 1
14:   **end if**
15:   **if** $\overline{H^{\text{sp}}} - 2\sigma^{\text{sp}} > H_i^{\text{sp}}$ and $\overline{H^{\text{ps}}} - 2\sigma^{\text{ps}} > H_i^{\text{ps}}$ **then**
16:    Traffic type $\leftarrow$ *Flooding Attack*
17:    **return** 0
18:   **end if**
19:   **if** $\overline{H^{\text{sp}}} + 2\sigma^{\text{sp}} < H_i^{\text{sp}}$ and $\overline{H^{\text{ps}}} - 2\sigma^{\text{ps}} > H_i^{\text{ps}}$ **then**
20:    Traffic type $\leftarrow$ *Packet-in Attack*
21:    **return** 0
22:   **end if**
23:  **else**
24:   Traffic type $\leftarrow$ *Unknown attack*
25:   **return** 0
26:  **end if**
27: **end for**

### 3.2.3. Proposed Detection Method

As shown in Algorithm 2, to determine if an attack is occurring, we first determine the average entropy $\overline{H^{di}}, \overline{H^{sp}}, \overline{H^{ps}}$ and the standard deviation $\sigma^{di}, \sigma^{sp}, \sigma^{ps}$ of each entropy in the window $\mathcal{W}_t \subset \mathcal{W}, t \in \mathcal{N}$ during which no attack is occurring.

Additionally, the entropy of each packet at the same window size follows an approximately normal distribution [27]. As is well-known, for variables that conform to a normal distribution, the data within one standard deviation accounts for 68.26% and within two standard deviations accounts for 95.45%. Therefore, this paper employs the mean value of the three types of entropy in the normal state at each time point, along with a system threshold of two times the standard deviation, to measure the degree of disorder for the three parameters at each time point. This approach minimizes errors that may arise from sampling randomness and effectively reduces the computational workload for each time point. The quantification is defined as follows:

- When the mean entropy value is within $\overline{H} \pm 2\sigma$, it is judged as normal.
- When the mean entropy value is smaller than $\overline{H} - 2\sigma$, it is judged as concentrated.
- When the mean entropy value is larger than $\overline{H} + 2\sigma$, it is judged as dispersive.

As shown in Figure 6. Using the three types of states defined, an attack decision is made on the basis of the characteristics of the abnormal traffic. Initially, anomaly-type methods are used to determine the range of normal traffic.

Therefore, as shown in Table 1, if $H_i^{di}$ in window $\mathcal{W}_i$, $i \in \mathcal{N}$ satisfies condition $\overline{H^{di}} + 2\sigma^{di} < H_i^{di}$, or $\overline{H^{di}} - 2\sigma^{di} \leq H_i^{di} \leq \overline{H^{di}} + 2\sigma^{di}$. According to existing method, if the target IP in unit time is not concentrated at a specific value, it is judged as normal traffic.

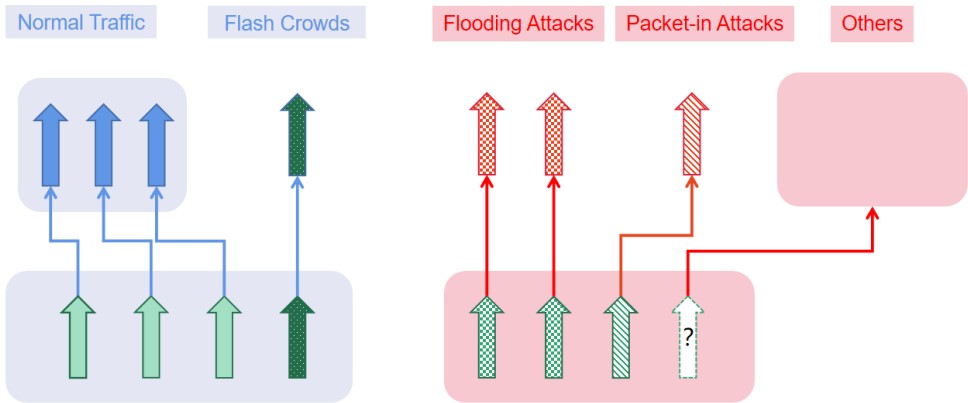

**Figure 6.** Proposed detection method.

**Table 1.** Abnormal traffic patterns.

|  | Flash Crowds | Flooding Attack | Packet-in Attack | Normal Traffic | Others |
|---|---|---|---|---|---|
| $H^{di}$ | 0 | 0 | 0 | 1 *or* 2 | - |
| $H^{sp}$ | 1 | 0 | 1 | - | - |
| $H^{ps}$ | 1 | 0 | 0 | - | - |
| Attack | 1 | 0 | 0 | 1 | 0 |
| Type | S | S | S | A | A |

To further reduce the false positive rate, we use a signature-type method to redefine similar anomalous traffic patterns that are outside the normal traffic range defined above.

First, we define the flash crowd [28], which are traffic patterns that are similar to attacks but are not attacks. A flash crowd is similar to a DDoS attack in that the traffic is concentrated on the server side; however, it is due to frequent accesses from users. Thus, this traffic can be judged as normal traffic. This means that there is little chance that the exact same packet exists in the flash crowd traffic, but the degree of variation in packet size is high. On the other hand, as the access frequency increases, the variability of the source port also increases. In summary, traffic that meets the three conditions of $\overline{H^{\mathrm{di}}} - 2\sigma^{\mathrm{di}} > H_i^{\mathrm{di}}$, $\overline{H^{\mathrm{sp}}} + 2\sigma^{\mathrm{sp}} < H_i^{\mathrm{sp}}$, $\overline{H^{\mathrm{ps}}} + 2\sigma^{\mathrm{ps}} < H_i^{\mathrm{ps}}$ will be determined as a flash crowd.

Flooding attacks, on the other hand, send the same packets with the goal of increasing the instantaneous traffic volume [29]. Therefore, the degree of dispersion of source ports and packet sizes is low. Unlike a flash crowd, traffic that meets the three conditions of $\overline{H^{\mathrm{di}}} - 2\sigma^{\mathrm{di}} > H_i^{\mathrm{di}}$, $\overline{H^{\mathrm{sp}}} - 2\sigma^{\mathrm{sp}} > H_i^{\mathrm{sp}}$, $\overline{H^{\mathrm{ps}}} - 2\sigma^{\mathrm{ps}} > H_i^{\mathrm{ps}}$ will be determined as a flooding attack.

The packet-in attack is an attack only in the SDN architecture that exploits the working principle of the OpenFlow protocol to send fake packets by sending a large number of request messages from the switch to the controller [30]. Since the size of the packet-in message is fixed, the degree of dispersion in packet size is low. On the other hand, since messages are sent from specific ports of each switch to the controller, the packet-in attack scale increases, and the degree of dispersion of the source port is higher than normal when the number of affected switches is large. In summary, traffic that meets the three conditions of $\overline{H^{\mathrm{di}}} - 2\sigma^{\mathrm{di}} > H_i^{\mathrm{di}}.\overline{H^{\mathrm{sp}}} + 2\sigma^{\mathrm{sp}} < H_i^{\mathrm{sp}}$, $\overline{H^{\mathrm{ps}}} - 2\sigma^{\mathrm{ps}} > H_i^{\mathrm{ps}}$ will be determined as a packet-in attack.

Finally, to take advantage of the anomaly-type method and to detect unknown attacks that have not been defined, we consider traffic patterns outside the normal traffic range that are not clearly defined as attacks [31]. The presence or absence of an attack (0 or 1) is determined according to the values shown in Table 1. Note that the entropy values of 1, 0, or 2 in the table refer to "$\overline{H} + 2\sigma < H_i$", "$\overline{H} - 2\sigma > H_i$", and "$\overline{H} - 2\sigma \leq H_i \leq \overline{H} + 2\sigma$", respectively, and the decision types *S* and *A* for each pattern refer to signature and anomaly types.

## 4. Numerical Analysis

### 4.1. Simulation Environment

In this simulation, as shown in Table 2, three typical types of anomalous traffic in the SDN architecture, ICMP flooding, flash crowds, and packet-in attacks, were simulated, and DDoS attack detection was performed using the proposed detection method on the parameters of packets collected during the simulation period. The SDN architecture used the Mininet software to construct the SDN net topology, and Open vSwitch (OVS) and Ryu controller were used for the SDN controller and OpenFlow switch. For this simulation environment, it was assumed that there were 12 hosts, 12 OVS switches, and 1 Ryu controller. A summary of the assumed environment is shown in the following Figure 7.

**Table 2.** Parameter settings.

| | Normal Traffic | Flash Crowds | ICMP Flooding | Packet-in Attack |
|---|---|---|---|---|
| Host number | 1–12 | 1–11 | 5–6 | — |
| Destination host number | 1–12 | 12 | 12 | — |
| Protocol | 85%TCP 10%UDP 5%ICMP | TCP | ICMP | UDP |
| Size | random | random | 42 | random |

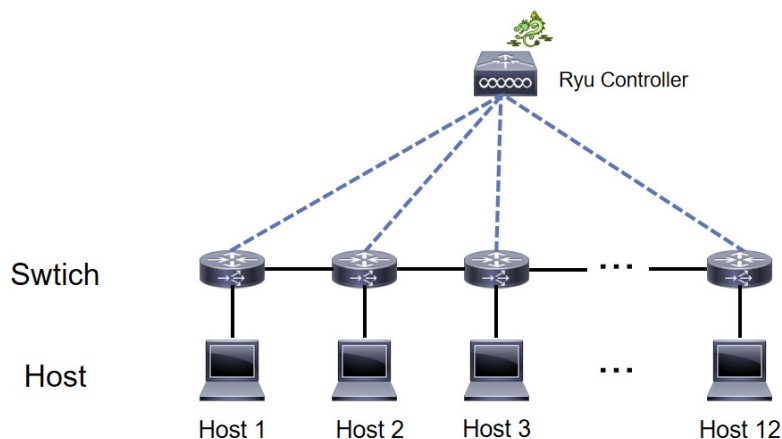

**Figure 7.** Network topology.

*4.2. Analysis and Evaluation*

In this paper, we used Scapy to generate normal traffic during $T = 150$ s and simulate abnormal traffic in the order of flash crowds, ICMP flooding, and packet-in attacks at regular intervals. In this simulation, we assumed the same conditions as in an existing study [9] and assumed that the amount of abnormal traffic was at least 5 times the amount of traffic arriving at all switches per unit time. To demonstrate the validity of the experiment, we also collected three datasets of attack intensity, 100%, 75%, and 50%, in order to perform entropy calculations by holding normal traffic to a constant standard and varying the frequency of attack traffic transmissions.

In this simulation, as shown in Figure 8:

- 1–25 s is normal traffic only
- 26–54 s is both normal traffic and flash crowd traffic
- 55–65 s is normal traffic only
- 66–82 s is both normal traffic and ICMP flooding traffic
- 83–112 s is normal traffic only
- 113–144 s is both normal traffic and packet-in attack traffic
- 145–150 s is normal traffic only

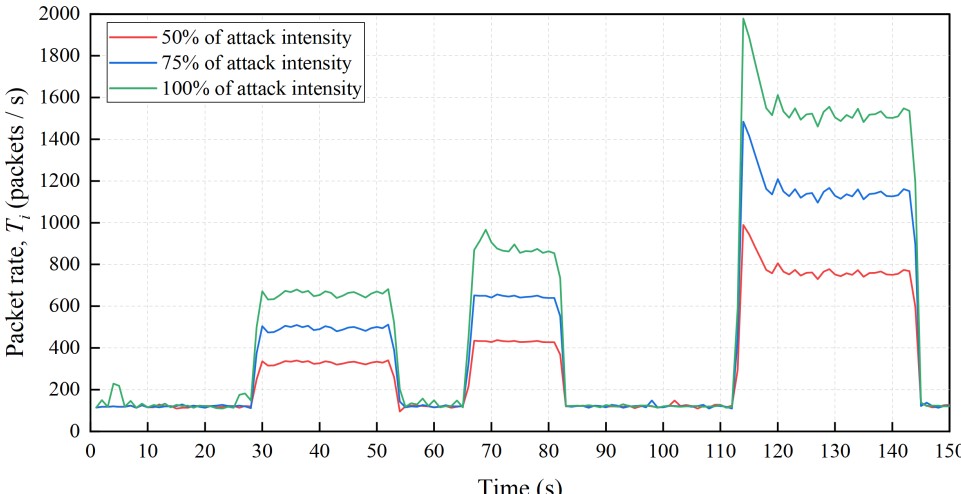

**Figure 8.** Amount of traffic.

In addition, As shown in Figure 9 specific normal traffic patterns and attack patterns were set as follows.

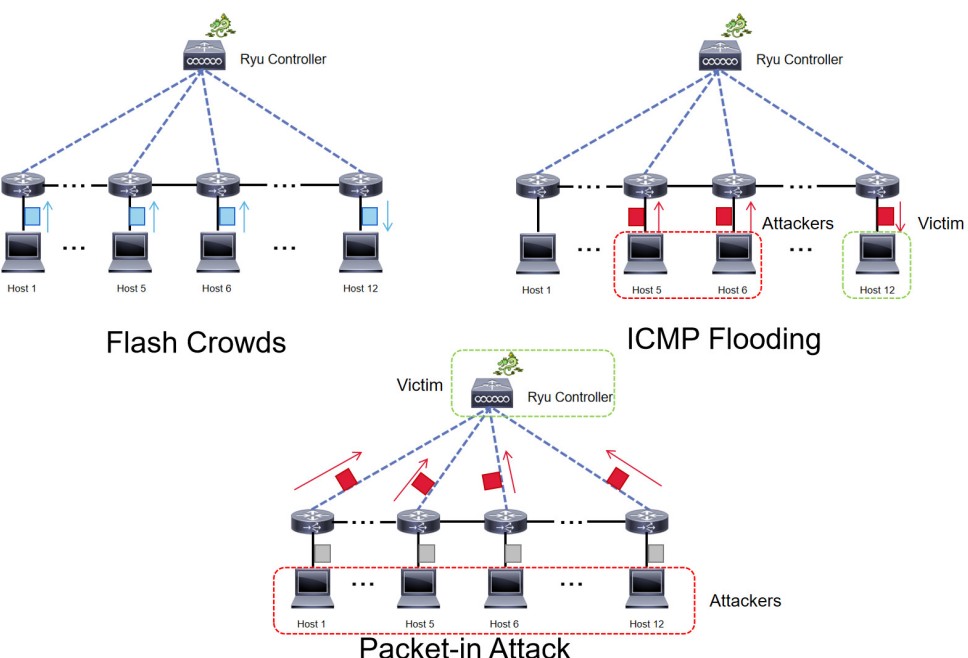

**Figure 9.** Attack patterns.

For numerical computation, we used Wireshark to analyse packets successfully sent from hosts in the SDN constructed during $T = 150$ s. We also computed entropy by selecting 100 pieces of data in each time interval, assuming a window size of $m = 100$.

The destination IP is a significant feature that can be leveraged for detecting DDoS attacks. As shown in Figure 10, during the period when the three types of abnormal traffic occurred, there was a substantial increase in the number of data packets sent to a specific destination IP address, resulting in a lower entropy for the destination IP than that of the normal state. Additionally, the number of transmissions to the same destination IP tended to increase. Therefore, the concentration of entropy for a single destination address is one of the conditions for determining the occurrence of any of the three types of abnormal traffic.

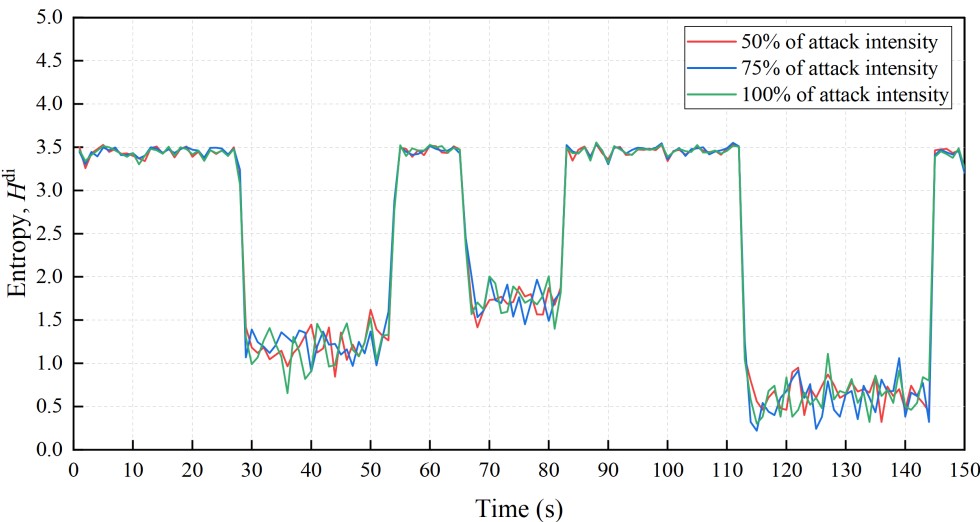

**Figure 10.** Destination IP entropy.

Figure 11 show the conditional entropy of the source port value change. Flash crowd occurrences are typically caused by network congestion resulting from frequent user requests for services. This, in turn, leads to a high frequency of source port changes,

resulting in an increased entropy of source ports based on the destination IP address compared to normal periods. During flooding attacks, attackers continuously send a large volume of packets to a specific destination address, leading to a drastic decrease in the entropy of source ports based on the destination IP address during the attack. In the case of packet-in attacks, attackers attempt to involve as many switches as possible in sending packet-in messages, leading to a slight increase in the entropy of source ports based on the destination IP address compared to normal periods.

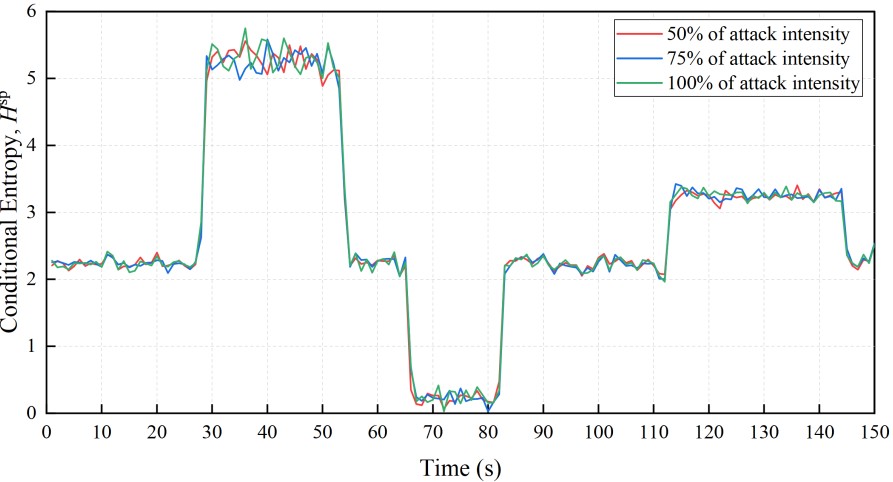

**Figure 11.** Source port conditional entropy.

Figure 12 show the conditional entropy of the packet size value change. In the case of flash crowds, actual user operations during this period make it nearly impossible for packet lengths to be completely consistent, resulting in an increased entropy of packet length based on the destination address. For flooding and packet-in attacks, the former requires a sufficiently high instantaneous flow rate, where attackers send a large number of data packets with the same length to conduct a flow attack. The latter is a result of the high volume of data packets generated in packet-in attacks based on the fixed length of the special message in the OpenFlow protocol. As a result, the entropy of packet length based on destination address will decrease during the occurrence of these two attacks.

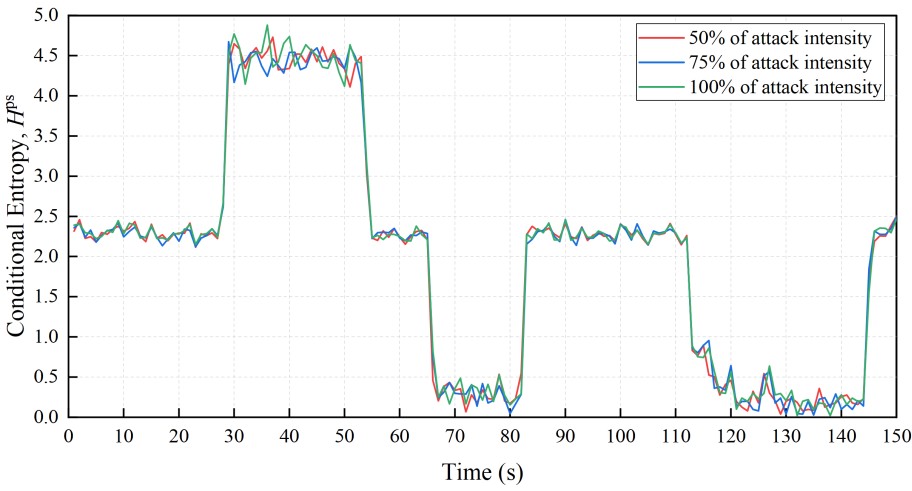

**Figure 12.** Packet size conditional entropy.

In this paper, precision, recall, *F*1 score, and accuracy were used as evaluation values [32]. By combining two of each detected and actual results, there were four patterns with correct and incorrect answers, called $TP$ (true positive), $FP$ (false positive), $TN$ (true negative), and $FN$ (false negative). The precision, recall, *F*1 score, and accuracy were calculated using the above $TP$, $FP$, $TN$, and $FN$ as follows. We used the confusion matrix to calculate the above performance indicators. The confusion matrix is shown in Figure 13.

$$Precision = \frac{TP}{TP + FP} \tag{7}$$

$$Recall = \frac{TP}{TP + FN} \tag{8}$$

$$F1Score = \frac{2 \times Precision \times Recall}{Precision + Recall} \tag{9}$$

$$Accuracy = \frac{TP + TN}{TP + FN + TN + FP} \tag{10}$$

**Figure 13.** Illustration of the confusion matrix.

We performed a comparative analysis of various entropy-based algorithms to demonstrate the efficacy of our proposed algorithm in achieving a balance between calculation accuracy and speed. Specifically, our proposed approach, along with [33,34], is based on conditional entropy, while [35] is based on joint entropy and [9,36] are based on single entropy. The comparison of the results is presented in Table 3.

**Table 3.** Performance of different methods

|  | Accuracy (%) | Precision (%) | Recall (%) | F1 Score (%) | Detecting Time (s) |
|---|---|---|---|---|---|
| Proposed method | 97.2 | 100 | 94.2 | 97.0 | 0.74 |
| [33] | 93.6 | 95.9 | 90.4 | 93.1 | 0.75 |
| [34] | 97.7 | 100 | 96.1 | 98.0 | 14 |
| [35] | 77.9 | 100 | 65.3 | 79.0 | 34 |
| [9] | 67.9 | 95.9 | 59.5 | 73.4 | 0.06 |
| [36] | 73.2 | 91.8 | 64.3 | 75.8 | 0.15 |

In contrast, As shown in Figure 14, the proposed conditional entropy-based method achieves the highest accuracy and recall scores while maintaining a high precision score. This is because the proposed method takes into account multiple parameters, including source and destination addresses, source and destination ports, and packet lengths, to evaluate the overall uncertainty of multiple parameters as a whole, which improves the detection accuracy and reduces false negatives. Therefore, the proposed method is a more effective and accurate method for DDoS attack detection compared to other entropy-based methods. Additionally, the proposed method in this paper also takes advantage of the feature of conditional entropy, which can measure the uncertainty of multiple parameters as a whole, instead of relying solely on single entropy or joint entropy. This approach provides a more comprehensive and accurate evaluation of abnormal traffic, especially for flash crowds, which can be challenging to detect using single entropy- or joint entropy-based methods. Overall, the proposed method achieves a balance between accuracy and efficiency, making it a promising approach for real-time DDoS detection.

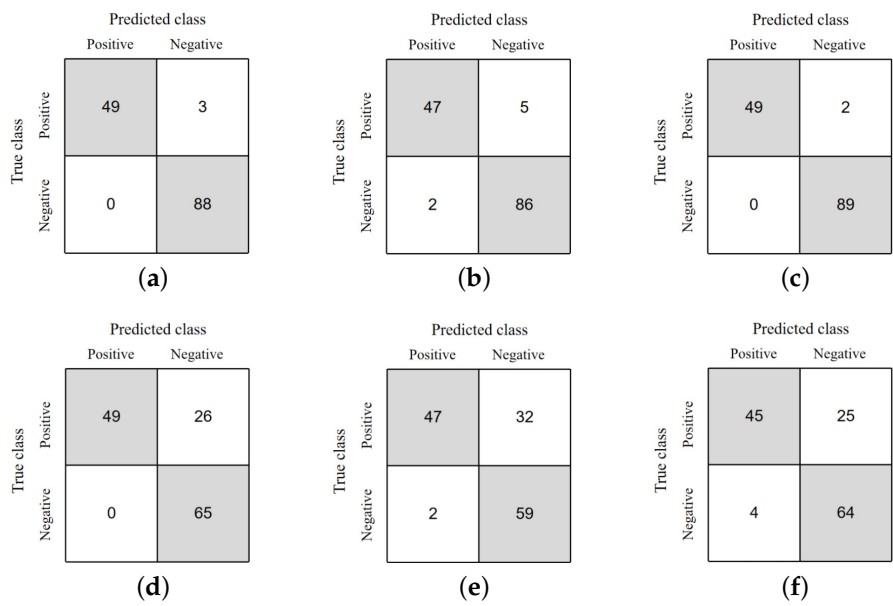

**Figure 14.** Comparison of the confusion matrix between six entropy methods. (**a**) Proposed method; (**b**) Conditional entropy [33]; (**c**) Conditional entropy [34]; (**d**) Joint Renyi entropy [35]; (**e**) Single entropy [9]; (**f**) Single entropy [36].

## 5. Conclusions

This paper proposes an attack detection method that uses conditional entropy to reduce the incidence of false positives by considering the similarity of abnormal traffic. Previous studies have used anomaly-type methods, which have two problems: insufficient attack patterns and unclearly defined degree of dispersion, leading to a high false positive rate in complex network environments. Therefore, this research defines attack patterns and normal traffic clearly by employing both signature- and anomaly-type methods. In addition, it individually defines parts that are similar to attacks and aggregates the range of normal traffic. Our proposed method for identifying anomalous traffic quantifies the concentration of traffic based on the mean and standard deviation, and uses changes in three types of entropy values to determine the type of traffic, thereby achieving more precise attack detection. Additionally, pre-processing is performed during traffic collection, so it is not necessary to traverse all collected packets, but only to process a random sample of packets to quickly obtain entropy values while maintaining a certain level of accuracy. Compared with existing methods, this approach has lower false positive rates, higher detection accuracy, and faster response times.

As a future challenge, the data collection part should be placed in the controller, and a system that notifies the switch as soon as traffic changes are detected in real time will enable higher-performance attack detection.

**Author Contributions:** Conceptualization, Q.T.; methodology, Q.T.; validation, Q.T. and S.M.; investigation, Q.T. and S.M.; resources, Q.T. and S.M.; writing—original draft preparation, Q.T.; writing—review and editing, Q.T. and S.M.; visualization, S.M.; project administration, S.M.; supervision, S.M.; funding, S.M. All authors have read and agreed to the published version of the manuscript.

**Funding:** These research results were obtained from the commissioned research (No. 05201) by the National Institute of Information and Communications Technology (NICT), Japan. This work was supported by JSPS KAKENHI Grant Numbers 19K11947, 22K12015, 22K12036.

**Institutional Review Board Statement:** Not applicable.

**Informed Consent Statement:** Not applicable.

**Data Availability Statement:** The datasets generated during the current study are available in the MDPI_den_2023, at https://github.com/sit-icnl/MDPI_den_2023 (accessed on 16 February 2023).

**Acknowledgments:** We would like to thank the anonymous reviewers for their comments and suggestions that help us improve the paper. Also We thank Zihan Zhang from University of Technology Sydney for his professional help in scripting.

**Conflicts of Interest:** The authors declare no conflict of interest.

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
