# Peer review of "A DDoS Attack Detection Method Using Conditional Entropy Based on SDN Traffic"

_2624-831X, doi:10.3390/iot4020006_

Round 1
Reviewer 1 Report
- The abstract of the paper should be revised and re-written appropriately to include the algorithms used in the model and the finding briefly.
- The motivation behind using conditional entropy to secure the SDN traffic should be discussed in the paper. Can the authors add more attacks other than DDoS?
- The paper needs English proofing, and it has typo mistakes.
- Enhance the quality of the figures (resolution, font size, etc...)
- Place the figures in the right order.
- Figure 4 is not informative.
- The work should be compared with more than one similar state-of-the-art models.
- The conclusion should be revised and rewritten appropriately to be consistent with the paper's argument and to summarize the model performance results.
- Most of the current references used in this paper are not state-of-the-art; please revise and add more state-of-the-art papers to enhance the literature, such as;
- Y. Otoum, Et al., "Federated and Transfer Learning-Empowered Intrusion Detection for IoT Applications," in IEEE Internet of Things Magazine, vol. 5, no. 3, pp. 50-54, September 2022.
- Chaganti, Rajasekhar, Wael Suliman, Vinayakumar Ravi, and Amit Dua. "Deep Learning Approach for SDN-Enabled Intrusion Detection System in IoT Networks." Information 14, no. 1 (2023): 41.
Reviewer 2 Report
The authors tried to implement a DDoS attack detection method that takes into account the packets in attacks in SDN environments. The IDS system is well explained in the article, but the “Analysis and evaluation” section needs improvement.
The following comments must be considered in the next version of this article. 1. A proper choice of threshold is crucial to a timely detection mechanism. The threshold setting is missing in the methodology. 2. Details regarding ‘Test Bed Environment ‘and which type ‘network topology’ selected is necessary. 3. Include output figures from the simulation itself instead of excel figures. 4. Table 3 and 4 can be merged to a single table. 5. Need summary of experimental results with various traffic scenarios. 6. Comparison of proposed result with reference [9] only shown in article, as there are a lot of recent research articles available based on “DDoS Attacks Using Conditional Entropy in SDN”. Comparison of proposed results with minimum five recent articles is relevant to prove the novelty. 7. Discuss the overall detection accuracy to detect DDoS attacks with the proposed system.Author Response
Please see the attachment

Round 2
Reviewer 2 Report
paper is fine now the authors handled all comments i raised last time.
Author Response
Thank you for your positive feedback on our revised manuscript.
We appreciate your efforts in providing constructive comments that helped us improve the quality of our work. We are glad to hear that you found our responses satisfactory and that you are satisfied with the current version of our paper. Your support and guidance have been invaluable in this process, and we look forward to continuing to work with you in the future.